# *Teucrium polium* (L.): Phytochemical Screening and Biological Activities at Different Phenological Stages

**DOI:** 10.3390/molecules27051561

**Published:** 2022-02-25

**Authors:** Majid Sharifi-Rad, Pawel Pohl, Francesco Epifano, Gokhan Zengin, Nidal Jaradat, Mohammed Messaoudi

**Affiliations:** 1Department of Range and Watershed Management, Faculty of Water and Soil, University of Zabol, Zabol 98613-35856, Iran; 2Department of Analytical Chemistry and Chemical Metallurgy, Faculty of Chemistry, University of Science and Technology, Wyspianskiego 27, 50-370 Wroclaw, Poland; 3Dipartimento di Farmacia, Università “Gabriele d’Annunzio” Chieti-Pescara, Via dei Vestini 31, 66100 Chieti Scalo, Italy; fepifano@unich.it; 4Physiology and Biochemistry Research Laboratory, Department of Biology, Science Faculty, Selcuk University, 42130 Konya, Turkey; gokhanzengin@selcuk.edu.tr; 5Department of Pharmacy, Faculty of Medicine and Health Sciences, An-Najah National University, Nablus P.O. Box. 7, Palestine; nidaljaradat@najah.edu; 6Nuclear Research Centre of Birine, P.O. Box 180, Ain Oussera, Djelfa 17200, Algeria; messaoudi2006@yahoo.fr; 7Chemistry Department, University of Hamma Lakhdar El-Oued, B.P. 789, El-Oued 39000, Algeria

**Keywords:** antioxidant activity, antibacterial activity, anti-inflammatory activity, phenological stages, *Teucrium polium*

## Abstract

The aim of the present study was to investigate the changes in the content of phytochemical compounds and in vitro antioxidant, antibacterial, and anti-inflammatory activities of *Teucrium polium* L. aerial parts and root methanolic extracts at different phenological stages (vegetative, flowering, and seeding). The *T. polium* extracts were analyzed using gas chromatography–mass spectrometry (GC-MS), and their antioxidant properties were tested with the 2,2-diphenyl-1-picrylhydrazyl (DPPH), nitric oxide (NO), ferrous ions (Fe^2+^), and 2,2′-azino-bis(3-ethylbenzothiazoline-6-sulfonic acid (ABTS) methods. Forty-nine compounds were identified with the majority of germacrene D, t-cadinol, β-pinene, carvacrol, bicyclogermacrene, α-pinene, and limonene. The results show that the extracts significantly differ between different phenological stages of the plant material used in terms of the phytochemical composition (total phenolic compounds, total flavonoids, total alkaloids, and total saponin contents) and bioactivities (antioxidant, antibacterial, and anti-inflammatory) (*p* < 0.05). The highest total contents of phenolics (72.4 ± 2.5 mg gallic acid equivalent (GAE)/g dry weight), flavonoids (36.2 ± 3.1 mg quercetin equivalent (QE)/g dry weight), alkaloids (105.7 ± 2.8 mg atropine equivalent (AE)/g dry weight), and saponins (653 ± 6.2 mg escin equivalent (EE)/g dry weight), as well as antioxidant, antibacterial, and anti-inflammatory activities, were measured for the extract of the aerial parts obtained at the flowering stage. The minimum inhibitory concentration (MIC) values for the extracts were varied within 9.4–300 µg/mL, while the minimum bactericidal concentration (MBC) values were varied within 18.75–600 µg/mL. In addition, they were more active on Gram-positive bacteria than Gram-negative bacteria. The data of this work confirm that the *T. polium* extracts have significant biological activity and hence can be used in the pharmaceutical industry, clinical applications, and medical research, as well as cosmetic and food industries.

## 1. Introduction

Plants are a significant source of bioactive constituents including phenols, aromatic components, terpenoids, sterols, essential oils, alkaloids, tannins, and anthocyanins that play a significant role in the treatment of many diseases [1]. The biological activity of plants depends on the amount and type of their constituents. In recent years, the biological activity of medicinal plants (antioxidant, anti-aging, anticancer, antifungal, anti-inflammatory, and antibacterial properties) has received special attention [2]. Polyphenolic compounds of medicinal plants have antioxidant activity in terms of their reducing power, free-radical scavenging property, and possibility to suppress the formation of singlet oxygen [3]. It is well known that antioxidant compounds have a protective role against many degenerative diseases and can be used as a dietary supplement. Moreover, functional food and food producers are interested in natural antioxidants necessary to prepare functional foods or to improve the nutritional quality of processed foods [4]. Many studies reported that various medicinal plants are valuable sources of various molecules with the antioxidant and antibacterial activity that can protect the human body against various pathogens as well as cellular oxidation reactions. These molecules are able to control and inhibit pathogens not causing any high toxicity to cells, and for that reason, certain medicinal plants can be recommended as appropriate materials for new antimicrobial agent studies [5]. Antibiotic resistance is an immediate threat to the treatment of infectious diseases; therefore, the search for new antimicrobial agents, especially from plant materials, is the main line of many studies aiming at counteracting this dangerous threat. Different phytochemicals show the potential antibacterial activity against resistant and sensitive pathogens, and therefore, they can be promisingly considered as candidates for the development of new drugs [6].

Inflammation is a non-specific response used by the innate immune system against pathogens, dangerous stimuli such as allergens, or tissue damages. Unregulated inflammatory responses are the main cause of several disorders, including cardiovascular disorders, allergies, tumors, metabolic syndromes, and autoimmune diseases [7]. To regulate different inflammatory crises, synthetic chemical drugs, including immunosuppressants and steroids or nonsteroidal anti-inflammatory drugs, are used; however, they have adverse effects associated with cardiovascular, gastrointestinal, and kidney functioning [8]. Alternatively, natural compounds and herbal medicines can be used as beneficial and increasingly important remedies for the prevention and treatment of inflammatory diseases. The anti-inflammatory activity of medicinal plants is related to the presence of natural antioxidants such as polyphenols, flavonoids, carotenoids, tocopherols, and ascorbic acid [9,10].

*Teucrium polium* L. (Lamiaceae family) is a perennial wild flowering plant, widely distributed in North Africa, Europe, and South-Western Asia. This species is considered as a significant ingredient in many traditional medicine prescriptions. *T. polium* is applied for different pathological conditions, including inflammations, gastrointestinal disorders, rheumatism, and diabetes in Iranian folk medicine [11]. Its tea is applied to treat various diseases, i.e., indigestion, common cold, abdominal pain, and urogenital diseases [12]. Many patients with type 2 diabetes, especially in Southern Iran, use the aqueous extracts obtained from *T. polium* aerial parts as an anti-diabetic drug [13]. The aqueous extract is also widely used in traditional medicine for treating stomach ulcers in some Arabian countries [11]. In Saudi Arabian traditional medicine, the infusions of the aerial parts *T. polium* and its tender leaves are applied by local peoples to treat vermifuge, stomach, febrifuge, and intestinal troubles. They are also utilized to treat fevers and colds using steam baths [14]. In Jordan, *T. polium* is used for treating various diseases such as diabetes, cancer, rheumatism, kidney stones, inflammation, pain, and fever [15]. In Turkish folk medicine, the infusions of its aerial parts are employed against eczema and hemorrhoids [16]. It is also used as a spice in meals. The aerial parts of *T. polium* are also brewed and drunk to treat stomach diseases [17]. The hard parts of this plant are boiled and then applied as a drug on the wounds [17]. *T. polium* is broadly used in the traditional medicine of North Africa. Its aerial parts are applied for the treatment of various diseases, i.e., liver problems, hypertension, fever, digestive disorders, rheumatism, inflammation, diabetes, and parasitic diseases in Moroccan folk medicine [18]. In Algeria, it is traditionally used due to its hypolipidemic, antioxidant, antibacterial, hypoglycemic, and anti-inflammatory properties [19]. Traditionally, Palestinians utilize its hot water leaf extracts for the treatment of cardiac and intestinal disorders, while the brewed leaves are drunk after each meal because of the anti-diarrheal and antispasmodic effects of such brews [20]. The crushed leaves are used on skin as a poultice for scabies [21]. In the Mediterranean countries, *T. polium* is traditionally used for several pathological conditions. It is utilized for gastrointestinal disorders in Bosnia and Herzegovina [22], for digestive problems in Albania [23], and as the anti-icteric, antihelmintic, and tonic product in Spain [24]. In Italy, it is applied against stomach pains, cold, myalgias, toothache pains, skin diseases, menopause disorders, and insect bites [25]. It is also recognized to produce advantageous tonic, diuretic, antifungal, antipyretic, anti-spasmodic, antibacterial, carminative, and antioxidant effects [26,27]. *T. polium* contains various classes of compounds, i.e., monoterpenes, diterpenes, fatty acid esters, sesquiterpenes, polyphenolics, and flavonoids [28]. It has been reported that *T. polium* compounds have anti-profilative, anti-diabetic, anticancer, and pro-apoptotic activities [15,29]. Recently, a lot of studies, focusing on the biological and pharmacological activity of plants in folk medicine, including the plant of *Teucrium polium* L., mostly have indicated significant differences in chemical composition (total flavonoids, total phenolic contents) and biological activity due to differences in the geographical location of the plant material but not differences due to its phenological stage [30,31].

For this purpose, the main objective of the present work was to assess, for the first time, the changes in the phytochemical composition of the methanolic extracts of the aerial parts and root of *T. polium* at its different phenological stages. In addition, the changes in the antioxidant, antibacterial, and anti-inflammatory activities of these extracts were investigated.

## 2. Results and Discussion

### 2.1. Total Phenolics Content

The results on the total content of phenolics of the *T. polium* aerial parts and root extracts, determined at different phenological stages (vegetative, flowering, and seeding) of this medicinal plant, are represented in Figure 1A. It was established that there were significant differences among the prepared extracts (*p* < 0.05). The highest total content of phenolics (72.4 ± 2.5 mg GAE/g dry weight) was found for the extract made of the aerial parts obtained at the flowering stage, while the lowest one (31.2 ± 1.3 mg GAE/g dry weight) was measured in the case of the root extract of the same phenological stage. In general, the biological activity of plants (and their extracts) is related to the presence of the compounds that have antioxidant and antimicrobial activity, with a special focus on phenols that have both of these properties [32]. The total content of phenolic compounds assessed for medicinal plants appears to be greatly affected by environmental factors and their phenological stages [33]. In this study, the *T. polium* plants were grown under the same conditions, although the samples used for the preparation of the aerial parts and root extracts were obtained at their different phenological stages. Hence, the differences observed for the total content of phenolic compounds were likely to be growth-stage-dependent. The results of this study confirm that the plant phenological stages play an important role in the formation and accumulation of certain phenolic compounds in the *T. polium* aerial parts and root and, in consequence, their total contents in the respective extracts.

### 2.2. Total Flavonoids Content

The results of the total content of flavonoids in the extracts of the *T. polium* aerial parts and root, collected at different phenological stages (vegetative, flowering, and seeding) of this medicinal plant, are shown in Figure 1B. This measure was also found to be varied due to the use of the plant material collected during different phenological stages. The extract of the aerial parts obtained at the flowering stage showed the highest value of the total flavonoids content (36.2 ± 3.1 mg QE/g dry weight) as compared to the other extracts under evaluation. Contrariwise, the extract of the *T. polium* root of the same phenological stage showed the lowest total flavonoids content among others (14.1 ± 1.2 mg QE/g dry weight). Flavonoids mainly accumulate in younger plants, and their content is reduced after the flowering stage when the plant is actively differentiating other than the synthesis of metabolites [34]. The reduction of the flavonoids content could be due to the fact that the plant uses these compounds for defense, pollination, and reproduction [35]. In addition, it was found in this study that the biosynthesis and the accumulation of flavonoids occur independently in each organ of *T. polium*, and their content varied depending on the plant growth stage [36,37].

### 2.3. Total Alkaloids Content

The results on the total content of alkaloids determined in the extracts of the *T. polium* aerial parts and root, gathered at different phenological stages of this plant, are presented in Figure 1C. Similar to the case of the total content of phenolics and flavonoids, the highest content of alkaloids (105.7 ± 2.8 mg AE/g dry weight) was determined in the extract made of the aerial parts of the plant collected at the flowering stage. As before, the lowest result (62.5 ± 1.6 mg AE/g dry weight) was noted for the root extract originating from the plant material of the same phenological stage. The other studies also report that the alkaloids levels change during different stages of the plant’s growth [38,39,40].

### 2.4. Total Saponins Content

As shown in Figure 1D, the highest total content of saponins (653 ± 6.2 mg EE/g dry weight) was also determined in the extract of the aerial parts of the plant at its flowering stage, while the lowest one (192 ± 5.3 mg EE/g dry weight) was measured for the root extract made of the material collected at the same phenological stage. The statistical analysis made for the total content of saponins of the extracts of *T. polium* prepared from the aerial parts and root of this plant, collected at different phenological stages, showed significant differences (*p* < 0.05). These results are also in accordance with the previous findings for *Cyclocarya paliurus* and *Chenopodium quinoa* [41,42]. In reference to the results of the current study, it was established that immature plants of *T. polium* contain higher concentrations of saponins as compared to mature plants. Several factors, i.e., environmental factors and physiological age, could affect the content of saponins in the plants [43].

### 2.5. Gas Chromatography–Mass Spectrometry Analysis

Using GC-MS, different volatile hydrocarbons were identified and determined in the *T. polium* extracts (see Table 1). The major compounds were germacrene D, t-cadinol, β-pinene, carvacrol, bicyclogermacrene, α-pinene, and limonene. The maximum and minimum concentrations of these compounds were observed in the extracts prepared from the aerial parts of the plant collected at the flowering stage and the root collected at the same stage, respectively, and this was coincident with the behavior of the total contents of phenolics, flavonoids, alkaloids, and saponins. Several studies report the antioxidant, antibacterial, and anti-inflammatory activity of these constituents [44,45,46]. Kerbouche et al. [47] analyzed the essential oils and ethanolic extracts of the *T. polium* plant, in which t-cadinol (18.3 %), germacrene D (15.3 %), and β-pinene (10.5 %) were the main compounds.

### 2.6. Antioxidant Activity

#### 2.6.1. DPPH Radical Scavenging Activity

The stable DPPH free-radical assay is widely applied to investigate the antiradical ability of different natural products [48]. The scavenging ability of the compounds toward DPPH is thought to be due to their hydrogen-donating ability [49]. The results on the radical scavenging potential of the extracts of the *T. polium* aerial parts and root originating from different phenological stages are given in Figure 2. As anticipated on the basis of the chemical analyses already carried out in this work, the extract of the aerial parts of the plant collected at the flowering stage had the highest DPPH radical scavenging activity, i.e., 83%. The lowest one was established for the extract of the root, which was collected at the same phenological stage, i.e., 26%. In the present study, as in other studies reporting the behavior of other plants [50,51,52,53], the DPPH radical scavenging activity determined in all analyzed *T. polium* extracts was established to be positively correlated with the total contents of phenolics, flavonoids, alkaloids, and saponins determined in these extracts. Accordingly, the respective Pearson’s correlation coefficients (r) were 0.984 (total phenolics), 0.863 (total flavonoids), 0.803 (total alkaloids), and 0.839 (total saponins) and indicated a very strong correlation of these measures as the r values were in the range from 0.8 to 1.0. Based on these results, the total contents of phenolics were mainly responsible for the antioxidant activity and highly affected by the position and number of the hydrogen-donating hydroxyl groups [54]. Many studies confirmed that phytochemical compounds such as phenolics, flavonoids, alkaloids, and saponins scavenge DPPH radicals by donating hydrogen atoms [55,56]. The results of this study suggest that the *T. polium* extracts had radical scavenging activity due to their hydrogen-donating or electron-transfer ability.

#### 2.6.2. ABTS Radical Scavenging Activity

A relatively stable ABTS radical is recommended for estimating the antioxidant activity of plant extracts, as the color of the plant extracts does not interfere with this estimation [57]. The reaction between ABTS and potassium persulfate produces the ABTS chromophore due to the conversion of ABTS to its radical cation [58]. ABTS cations are reactive to the most plant-based antioxidant agents, i.e., phenolic and flavonoids compounds [59]. The results of the ABTS radical scavenging activity of the extracts of the *T. polium* aerial parts and root collected at different phenological stages are presented in Figure 3. As can be seen, the phenological stages and plant parts had a significant effect on the ABTS radical scavenging activity (*p* < 0.05) of the prepared extracts. The measured ABTS radical scavenging activity of these extracts was decreased in the following order: the aerial parts taken at the flowering stage (78%) > the aerial parts taken at the vegetative stage (76%) > the root taken at the seeding stage (42%) > the root taken at the vegetative stage (34%) > the aerial parts taken at the seeding stage (28%) > the root taken at the flowering stage (15%). In addition, there were found very strong positive correlations between the ABTS radical scavenging activity and the content of total phenolics (r = 0.979), flavonoids (r = 0.905), alkaloids (r = 0.862), and saponins (r = 0.844). These results are in accordance with the previous studies reported for other plants [60,61,62]. Extensive research reports on the antiradical and antioxidant activities of small phenolics such as phenolic acids, flavonoids, and alkaloids [53,63]. In addition, it is revealed that the phenolics compounds with the higher molecular weight (saponins) have a higher potential for quenching the ABTS radicals and that their effectiveness depends on the aromatic ring number, nature of the replacement of hydroxyl groups rather than particular functional groups, and molecular weight [64].

#### 2.6.3. Nitric Oxide Radical Scavenging Activity

The nitric oxide (NO) free radical is produced in the cells and involved in regulating different physiological processes. However, the overproduction of NO leads to several diseases [65]. The production of a stable level of this radical has a toxic effect on tissues and leads to a vascular collapse related to the septic shock, while its chronic expression is related to various cancers and inflammatory conditions, i.e., multiple sclerosis, arthritis, juvenile diabetes, and ulcerative colitis [66]. When NO reacts with the superoxide radical, the highly reactive peroxynitrite anion (ONOO^−^) is formed, and thus, the NO toxicity is greatly increased [67]. The results of the NO radical scavenging activity of the extracts of the *T. polium* aerial parts and root collected at different phenological stages are given in Figure 4. In addition, the NO radical scavenging activity of butylated hydroxyanisole (BHA) is included for comparison purposes. The maximum NO radical inhibition percentage of 52% was measured in the extract of the aerial parts collected at the flowering stage. It was followed by the values of 45% determined in the extract of the aerial parts collected at the vegetative stage, and of 43% determined for the extract of the root collected at the seeding stage. The 2–3-times lower values of the NO radical inhibition percentages were acquired for the other extracts. All the NO radical inhibition percentages determined in the extracts of the *T. polium* plant material were lower than that assessed for the BHA standard (64%) at the same concentration. A strong positive correlation was only found between the NO radical scavenging activity and the total content of phenolics (r = 0.957). The correlations between the NO radical scavenging activity and the total contents of flavonoids, alkaloids, and saponins were positive and strong because the r values were 0.695, 0.638, and 0.699, respectively. NO free radical is produced due to the interaction of NO molecules with O_2_ molecules. The phenolic compounds, as antioxidant agents, can compete with O_2_ to combine with NO molecules and lead to a significant decrease in the formation of the free radicals due to the conversion of NO to its reducing products [68]. The current study confirmed that the studied extracts had the appropriate potential for the NO scavenging activity.

#### 2.6.4. Ferrous Ion Chelating Activity

Fe^2+^ ions have the ability to continue the generation of free radicals by the loss and gain of electrons. The formation of these radicals can cause protein modification, lipid peroxidation, and DNA damage. The chelating agents are able to inactivate transition metal ions and prevent the formation of the reactive oxygen species (ROS) [69].

The results on the ferrous ion chelating activity of the extracts of the *T. polium* aerial parts and root, gathered at different phenological stages, are presented in Figure 5. It can be seen that all the *T. polium* extracts were able to chelate the ferrous ions. The extract made of the aerial parts of the plant material possessed at the flowering stage indicated the highest ferrous ion chelating activity of 32%. The lowest one of 11% was assessed for the extract of the root collected at the same phenological stage. The positive correlation between the ferrous ion chelating activity and the total contents of phenolics was very strong (r = 0.696), indicating that this class of compounds mainly contributed to this measure. In the case of the ferrous ion chelating activity and the total contents of flavonoids, alkaloids, and saponins, the positive correlations found between them were strong (r = 0.774, r = 0.732, and r = 0.696, respectively). Other studies also expressed that there is a positive correlation between the ferrous ion chelating activity and the amount of phytochemical compounds such as phenolics, flavonoids, alkaloids, and saponins present in the extracts prepared from other plants [70,71,72].

### 2.7. Antibacterial Activity

#### 2.7.1. Disc Diffusion Method

The antibacterial activity of the extracts of the *T. polium* aerial parts and root collected at different phenological stages was investigated on Gram-positive and Gram-negative bacteria and using the agar disk diffusion method. Figure 6 shows the inhibition zone diameters (mm). It can be seen that all the examined extracts of *T. polium* exhibited an appropriate antibacterial activity on the studied bacteria strains. It was found that these extracts had a higher biocidal activity against the Gram-positive bacteria than the Gram-negative bacteria strains. This is likely due to the cell wall structure of the Gram-positive bacteria, which is simpler than that of the Gram-negative bacteria. The Gram-negative bacteria have rigid cell walls, the main component of which are lipopolysaccharides, making these bacteria more resistant to different antibacterial agents. In contrast, the Gram-positive bacteria cell walls are homogeneous and do not contain phospholipids [73]. It is reported that the plant extracts are hydrophobic in nature, while the Gram-negative bacteria outer membranes have some hydrophilic porins; hence, the diffusion of the plant extract compounds into the Gram-negative bacteria is hindered [74].

#### 2.7.2. Minimum Inhibitory Concentration (MIC) and Minimum Bactericidal Concentration (MBC)

The results on the MIC and MBC values measured for the extracts of the *T. polium* aerial parts and root collected at different phenological stages are presented in Table 2. In line with the results of the disc diffusion assay, the micro-dilution test results approve that the Gram-positive bacteria were more sensitive to the different extracts of the *T. polium* plant material than the Gram-negative bacteria strains. The MIC values of the different extracts of the *T. polium* plants for the tested pathogenic microorganisms varied from 9.4 to 300 µg/mL, while the MBC values ranged from 18.75 to 600 µg/mL.

### 2.8. Anti-Inflammatory Activity

#### Human Red Blood Cell Stabilization

The anti-inflammatory activity results evaluated for the extracts of the *T. polium* aerial parts and root plant material collected at different phenological stages are presented in Figure 7. As can be seen, the different extracts of the *T. polium* plant material showed the appropriate stabilizing properties on the human red blood cell membranes. The extracts of the aerial parts of this plant, obtained using the material gathered at the flowering stage, indicated the highest anti-inflammatory activity of 68.5%. On the contrary, the extract of the root of this plant, obtained using the material gathered at the flowering stage, showed the lowest anti-inflammatory activity of 19.4%. The HRBC assay was used for the evaluation of the anti-inflammatory activity because the red blood cell membranes are similar to the lysosomal membranes [75] and their stabilization means that the extracts of the *T. polium* plant material may also stabilize the lysosomal membranes. In the process of inflammation, the lysosomal constituents enter the cytosol and cause various disorders by damaging the adjacent tissues [76]. It is suggested that the plant extracts might prevent the release of the lysosomal compounds from the neutrophils at the inflammation site [77]. The outcomes of the present work revealed that there were very strong positive correlations between the anti-inflammatory activity and the total contents of phenolics (r = 0.884), flavonoids (r = 0.979), alkaloids (r = 0.966), and saponins (r = 0.916). These results are in accordance with the previous studies reported for other plants [78,79,80]. These phytochemical substances were already proven to possess anti-inflammatory properties [81]. Among them, terpenes, alkaloids, and phenolic compounds, i.e., flavonoids, lignans, saponins, tannins, and coumarins, are especially prominent [82].

## 3. Materials and Methods

### 3.1. The Plant Extract Preparation

The aerial parts and root of *T. polium* L. were collected at different phenological stages from the Saravan rangelands, Sistan, and Baluchestan, Iran (27°17′30″ N, 62°17′11″ E). The species was identified at the Department of Range and Watershed Management, University of Zabol, Iran, and a voucher specimen of the plant (No. 1154) was deposited for future reference. The plant samples were dried at 40 °C using an oven for 72 h and finally ground by using an electric grinder (Pars Khazar, Tehran, Iran) to obtain a fine powder. The extraction was performed magnetically stirring 5 g of the powder with 50 mL of methanol for approximately 10 h at room temperature (25 ± 1 °C). The higher extraction capacity of methanol can produce a large number of active compounds responsible for the biological activity of the prepared extracts [83,84]. The extracts were filtered using Whatman filter papers No. 1. The resulted extracts were evaporated under vacuum conditions to dryness and stored for the next experiments at 4 °C.

### 3.2. Phytochemical Analysis

#### 3.2.1. The Quantification of the Total Contents of Phenolics

The total content of phenolics was measured in the different extracts of the *T. polium* plant material by using the spectrophotometric method [85]. Various extracts were prepared at the same concentration of 1 mg/mL for analysis. Briefly, 0.5 mL of each extract, 2.5 mL of the Folin–Ciocalteu’s reagent (10%) dissolved in water, and 2.5 mL of a NaHCO_3_ solution (7.5%) were mixed, and the reaction mixtures were incubated for 45 min at room temperature in dark. The absorbance of the resulting reaction mixtures was measured at 765 nm by a spectrophotometer (UV-1800 240 V, Shimadzu Corporation, Kyoto, Japan). The standard solutions of gallic acid were prepared and used to obtain the standard calibration curve (10–100 mg/mL, y = 0.008x − 0.0048, R^2^ = 0.997). The total content of phenolics in the examined samples was expressed as the gallic acid equivalent (mg of GAE/g dry weight).

#### 3.2.2. The Quantification of Total Content of Flavonoids

The total content of flavonoids of the different extracts of the *T. polium* plant material was determined according to the colorimetric assay described by Sharifi-Rad et al. [86] with some modifications. The extracts were prepared at the same concentration of 1 mg/mL for analysis. In summary, 0.5 mL of each extract, 0.1 mL of an AlCl_3_ solution (10%), 0.1 mL of a potassium acetate solution (1 mol/L), and 4.3 mL of distilled water were mixed, and the reaction mixtures were incubated for 30 min at room temperature. The absorbance of the resulting reaction mixtures was measured at 510 nm by the spectrophotometer (UV-1800 240 V, Shimadzu Corporation, Kyoto, Japan). Quercetin was applied for preparing the standard calibration curve (10–100 mg/mL, y = 0.0093x − 0.0413, R^2^ = 0.994). The total content of flavonoids of the examined samples was expressed as the quercetin equivalent (mg QE/g dry weight).

#### 3.2.3. The Quantification of Total Content of Alkaloids

The total content of alkaloids was determined by the colorimetric method described by Ajanal et al. [87] with minor modifications. In brief, the portions of the extracts of the *T. polium* material plant (1 mg/mL) were dissolved in dimethylsulphoxide (DMSO) and a 2 mol/L HCl solution (1 mL) and filtered. The resultant mixtures were transferred to separating funnels. Then, a phosphate buffer (5 mL) and a bromocresol green solution (5 mL) were added. The mixtures were vigorously shaken with chloroform and collected into 10 mL volumetric flasks. The absorbance of the resultant mixtures was measured at 470 nm by the spectrophotometer (UV-1800 240 V, Shimadzu Corporation, Kyoto, Japan). The standard solutions of atropine were prepared in a similar way and used to obtain the standard calibration curve (10–100 mg/mL, y = 0.0084x − 0.001, R^2^ = 0.9927). The total content of alkaloids of the examined samples was represented as the atropine equivalent (mg AE/g dry weight).

#### 3.2.4. The Quantification of the Total Content of Saponins

The total content of saponins of the different extracts of the *T. polium* material plant was evaluated as previously reported by Vuong et al. [88] with some modifications. In brief, 0.5 mL of the different extracts of the *T. polium* plant material (1 mg/mL) was mixed with 0.5 mL of a vanillin solution (8%), and then 5 mL of a H_2_SO_4_ solution (72%) was added to the mixture. The resulting mixtures were thoroughly mixed and placed on ice to cool. Then, they were incubated in a water bath for 15 min at 60 °C. Afterward, these mixtures were cooled again on ice, and their absorbance was measured at 560 nm by the spectrophotometer (UV-1800 240 V, Shimadzu Corporation, Kyoto, Japan). Escin was applied as a standard for preparing the calibration curve (10–100 mg/mL, y = 0.0083x + 0.0145, R^2^ = 0.9901). The total content of saponins was expressed as the escin equivalent (mg EE/g dry weight).

#### 3.2.5. Gas Chromatography–Mass Spectrometry Analysis

The gas chromatography–mass spectrometry analysis was performed using a GCMS-QP2010 system (Shimadzu, Tokyo, Japan). A total of 20 μL of each extract was diluted using hexane (≥99%, Sigma–Aldrich, Darmstadt, Germany) to 1 mL. The applied column was RTX-5MS (Restek, Bellefonte, PA, USA) (30 m × 0.25 mm i.d. × 0.25 µL film thickness). Helium (99.999%, AGA, Vilnius, Lithuania) was applied as carrier gas and passed at a flow rate of 1.23 mL/min. The temperature of the oven was held for 2 min at 40 °C after injection. Afterward, it was programmed to increase at 3 °C/min to 210 °C, and this temperature was held for 10 min. The split ratio was 1:10. The detection was carried out by 70-eV electron ionization. The identification of the volatile compounds was performed using the mass spectra library search (NIST 14) and compared with the mass spectral data from the literature [89].

### 3.3. Antioxidant Activity

#### 3.3.1. 2,2-Diphenyl-1-picrylhydrazyl (DPPH) Radical Scavenging Activity Method

The DPPH radical scavenging activity of the different extracts of the *T. polium* plant material was represented as the DPPH radical inhibition percentage and determined based on the assay reported by Sharifi-Rad and Pohl [90] with minor modifications. In brief, 1 mL of the different *T. polium* extracts (200 µg/mL) was mixed with 2 mL of a DPPH methanolic solution (1 mmol/L). The resulting solutions were completely mixed and incubated at room temperature (25 ± 1 °C) in dark for 30 min. Then, the absorbance of each mixture was measured at 517 nm using the spectrophotometer (UV-1800 240 V, Shimadzu Corporation, Kyoto, Japan). A BHA solution at a similar concentration was applied as a standard. The working solution of the DPPH radical was considered as the control. The DPPH radical scavenging activity (%) was estimated by the following equation: DPPH radical scavenging (%) = [(A_0_ − A_1_)/A_0_] × 100, where A_0_ is the absorbance of the control, and A_1_ is the absorbance of the different extracts of the *T. polium* plant material or the BHA solution.

#### 3.3.2. 2,2′-Azinobis(3-ethylbenzothiazoline-6-sulfonic Acid) (ABTS) Radical Scavenging Activity Method

The ABTS radical scavenging activity of the different extracts of the *T. polium* plant material was represented as the ABTS radical inhibition percentage and determined based on the assay reported by Ko et al. [91] with some modifications. A stock solution was prepared by mixing the equal volumes of an ABTS radical solution (7.0 mM) and a potassium persulfate (K_2_S_2_O_8_) solution (2.45 mM) and next kept in dark at room temperature (25 ± 1 °C) for 15 h to obtain a dark-colored solution containing the ABTS radical cations. The resulting solution was diluted in methanol (50%) to obtain the working solution of the ABTS radical with an initial absorbance of 0.70 ± 0.01 at 734 nm. A total of 300 μL of the different extracts of the *T. polium* plant material (200 µg/mL) was mixed with 3 mL of the ABTS radical working solution in a micro cuvette. After 30 min, the absorbance of the resulting mixtures was measured at 734 nm using the spectrophotometer (UV-1800 240 V, Shimadzu Corporation, Kyoto, Japan). The BHA solution at a similar concentration was applied as a standard. The working solution of the ABTS radical was considered as the control. The inhibition of the ABTS radical (%) was calculated using a similar equation as for the DPPH method.

#### 3.3.3. Nitric Oxide (NO) Radical Scavenging Activity

The NO radical scavenging activity of the different extracts of the *T. polium* plant material was determined according to the assay reported by Kamble et al. [92] with minor modifications. This assay is based on the inhibition of the NO radical generated from sodium nitroprusside, which is measured by the Griess reaction. The Griess reagent was obtained by mixing equal amounts of a 0.2% naphthylethylene diamine dihydrochloride solution in 4% H_3_PO_4_ and a 2% sulphanilamide solution in 4% H_3_PO_4_. A total of 0.5 mL of a sodium nitroprusside solution (10 mM) in the phosphate-buffered saline was mixed with 1 mL of the different extracts of the *T. polium* plant material (200 µg/mL) and incubated at room temperature (25 ± 1 °C) for 180 min. Then, an equal volume of the Griess reagent was added, and the resulting mixtures were incubated at room temperature for 5 min. The absorbance of each mixture was measured at 546 nm using the spectrophotometer (UV-1800 240 V, Shimadzu Corporation, Kyoto, Japan). The BHA solution at a similar concentration was used for comparison. The inhibition of the NO radical (%) was calculated in the same way as in the case of the DPPH method.

#### 3.3.4. Ferrous Ion (Fe^2+^) Chelating Activity

The ferrous ion (Fe^2+^) chelating activity of the different extracts of the *T. polium* plant material was represented as the inhibition percentage in the Fe^2+^-ferrozine complex formation and determined based on the method reported by Dinis et al. [93]. Briefly, 0.4 mL of the different extracts of the *T. polium* plant material (200 µg/mL) was added to 0.05 mL of a 2 mM ferrous chloride (FeCl_2_) solution. By the addition of 0.2 mL of a ferrozine solution (5 mM), the reaction was initiated. The resulting mixtures were shaken vigorously and kept at room temperature (25 ± 1 °C) for 15 min. Then, the absorbance of each mixture was measured at 562 nm using the spectrophotometer (UV-1800 240 V, Shimadzu Corporation, Kyoto, Japan). The BHA solution at a similar concentration was applied for comparison. The ferrous ion (Fe^2+^) chelating activity (%) was calculated using an equation similar to that given in the DPPH assay.

### 3.4. Antibacterial Activity

The antibacterial activity of the different extracts of the *T. polium* material plant was investigated using Gram-positive (*Bacillus cereus* (ATCC 8035) and *Staphylococcus aureus* (ATCC 25923)) bacteria strains and Gram-negative (*Shigella flexneri* (ATCC 12022) and *Escherichia coli* (ATCC 25922)) bacteria strains. The bacteria strains were obtained from the Iranian Research Organization for Science and Technology (IROST).

#### 3.4.1. Disc Diffusion Assay

The disc diffusion test was used to investigate the antibacterial activity of the different extracts of the *T. polium* plant material as reported by Sharifi-Rad et al. [94]. The sterile filter paper discs (6 mm in diameter) were impregnated with 30 µL of the different extracts of the *T. polium* plant material (200 µg/mL) and air-dried. A total of 100 µL of the 0.5 McFarland bacteria suspensions (containing 1.5 × 10^8^ CFU/mL of the bacteria strains) was dispersed on Müller–Hinton agar (MHA) plates. The paper disks were placed on the surface of the MHA plates at an appropriate distance from each other. Solvent-impregnated discs were considered as negative controls, and gentamicin 10 µg disks were applied as positive controls. All the plates were incubated at 37 °C for 24 h. After that, the antibacterial activity of the extracts was determined by measuring the clear zones of inhibition to the nearest millimeter (mm).

#### 3.4.2. The Minimum Inhibitory Concentration (MIC) Determination

The micro-broth dilution assay was applied to estimate the minimum inhibitory concentration (MIC) for the different extracts of the *T. polium* plant material on the studied pathogenic microorganisms as suggested by the Clinical and Laboratory Standards Institute [95]. The concentrations of the extracts used for the MIC determination were varied from 600 to 4.7 µg mL^−1^. The test was performed using polystyrene 96-well plates. A total of 50 µL of the different extracts of the *T. polium* plant material and 50 µL of the Müller–Hinton broth were poured into each well. Then, 50 µL of a 0.5 McFarland bacteria suspension was added to the wells. The plates were incubated at 37 °C for 24 h. The pure medium and the medium including the bacteria were considered as negative and positive controls, respectively. The lowest concentration of the different extracts of the *T. polium* plant material that showed no observable growth of the tested bacteria were intended as the MIC.

#### 3.4.3. The Minimum Bactericidal Concentration (MBC) Determination

The minimum bactericidal concentration (MBC) was determined according to the assay described by the Clinical and Laboratory Standards Institute [95]. A total of 50 µL of each well of the broth micro-dilution test that showed no observable bacterial growth was sub-cultured on MHA plates and incubated at 37 °C for 24 h. The lowest concentrations of the different extracts of the *T. polium* plant material that displayed no bacterial growth were intended as the MBC.

### 3.5. Anti-Inflammatory Activity

#### Human Red Blood Cell Stabilization Assay

The human red blood cell (HRBC) membrane stabilization method was used to evaluate the in vitro anti-inflammatory activity of the different extracts of the *T. polium* plant material as reported by Vane and Botting [96]. The blood samples were obtained from 15 healthy human volunteers and blended with an equal volume of an Alsever’s solution (consisting of 0.8% sodium citrate, 2% dextrose, 0.42% sodium chloride, and 0.5 % citric acid). The blood samples were centrifuged at 3000 rpm for 10 min, and the packed cells were separated. The resulting packed cells were washed with a 0.9% isosaline solution, and finally, a 10% *v/v* cell suspension was prepared in isosaline. This suspension was applied for the determination of the anti-inflammatory property of the different extracts of the *T. polium* plant material. According to this, 1 mL of the extracts (200 µg/mL) and a diclofenac sodium solution (as a reference drug) at the same concentration were separately mixed with 1mL of a phosphate buffer (0.15 mol L^−1^, pH 7.4), 2 mL of hyposaline (0.36%), and 0.5 mL of the HRBC suspension. A total of 2 mL of distilled water was used as the control. All the obtained mixtures were incubated at 37 °C for 30 min and centrifuged at 4000 rpm for 20 min. The resultant supernatants were evacuated and the hemoglobin content was determined using the spectrophotometer at 560 nm. The percentage of the hemolysis produced in the control was assumed to be 100%. The human RBC membrane stabilization (%) or the protection (%) was calculated by the following equation:Protection (%) = 100 − (OD _sample_/OD _control_) × 100

### 3.6. Statistical Analysis

The statistical analyses were carried out using the SPSS software (Version 11.5, SPSS Inc., Chicago, IL, USA). The analysis of variance (ANOVA) followed by Duncan’s multiple range test (DMRT) was applied at the 95% confidence level (α = 0.05). All of the measurements were carried out in triplicate. The results were represented as mean values ± SDs.

## 4. Conclusions

This study reported the phytochemical analysis of the aerial parts and roots of *T. polium* extracts collected at different phenological stages. In this work, the analysis of the extracts showed that the *T. polium* extracts have the appropriate antioxidant, antibacterial, and anti-inflammatory activity, the highest being for the aerial parts originated from the flowering stage. In addition, this extract indicated the highest amounts of different phytochemical compounds (phenolics, flavonoids, alkaloids, and saponins). As it is known and proven by a large number of researchers, these phytochemical components are responsible for the bioactivity of the extract.

The best conclusion for this research is that, to the best of our knowledge, this study is the first to report on the antioxidant, antibacterial, and anti-inflammatory activities of the extracts of *T. polium* aerial parts and root plant material collected at different phenological stages. These results support the use of this plant in folk medicine as a source of bioactive molecules acting on several human disorders and potentially useful as biologically active agents in food and pharmaceutical formulations.

## Figures and Tables

**Figure 1 molecules-27-01561-f001:**
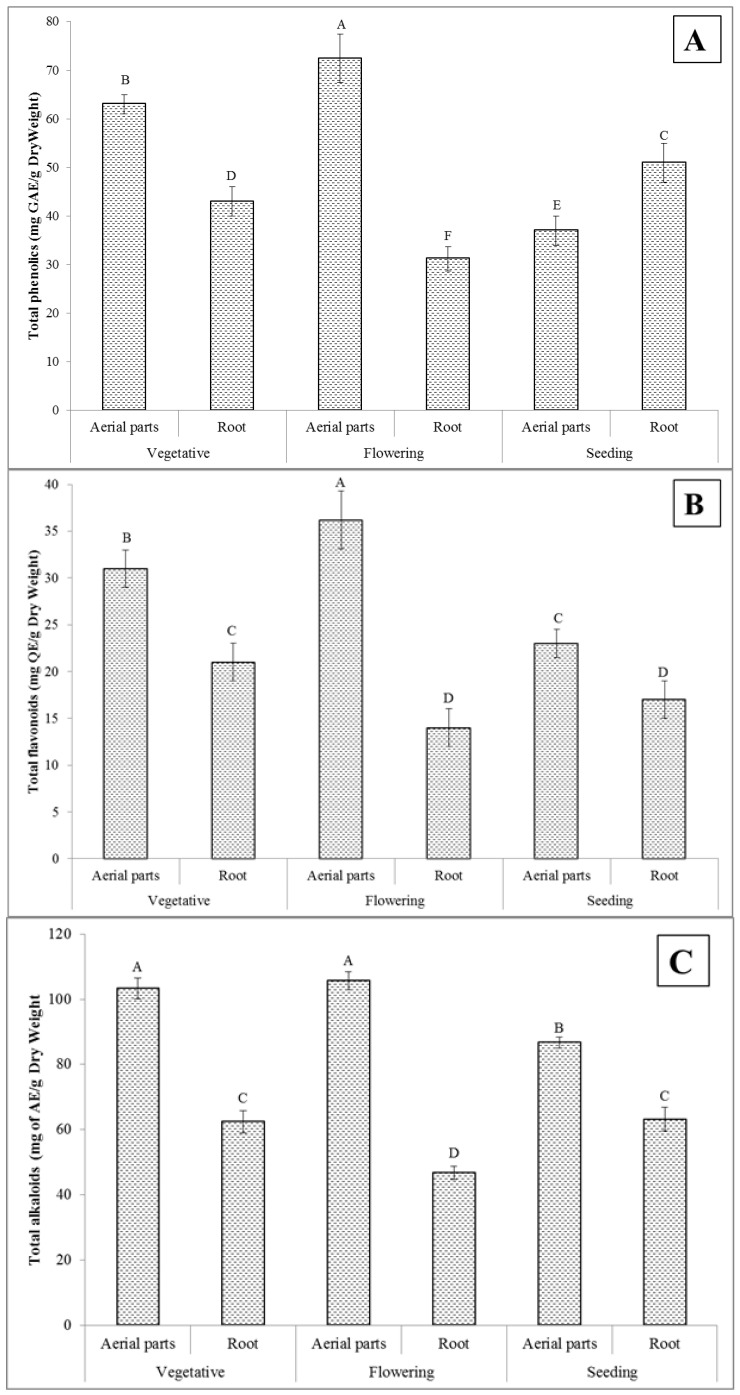
The total contents (*n* = 3) of (**A**) phenolics, (**B**) flavonoids, (**C**) alkaloids, and (**D**) saponins in the extracts of the *T. polium* aerial parts and root at different phenological stages of this plant. The extract concentration is 1 mg/mL. Various letters indicate statistically significant differences (*p* < 0.05).

**Figure 2 molecules-27-01561-f002:**
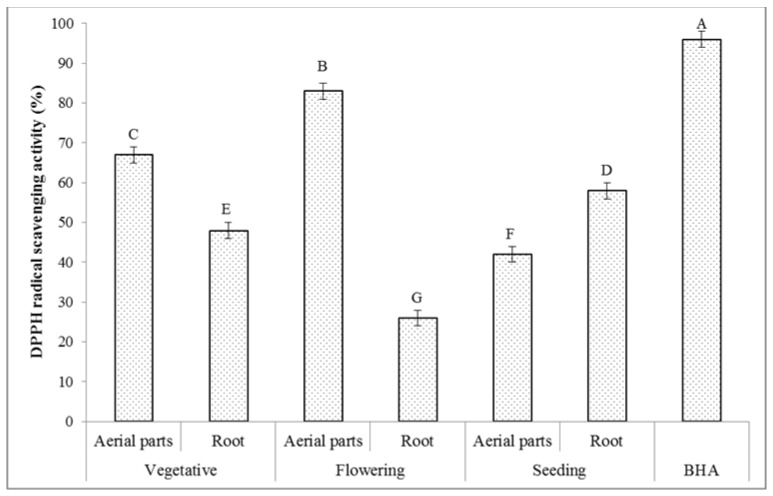
DPPH radical scavenging activity (*n* = 3) of the extracts of the *T. polium* aerial parts and root at different phenological stages. The extract concentration is 200 µg/mL. Various letters indicate statistically significant differences (*p* < 0.05).

**Figure 3 molecules-27-01561-f003:**
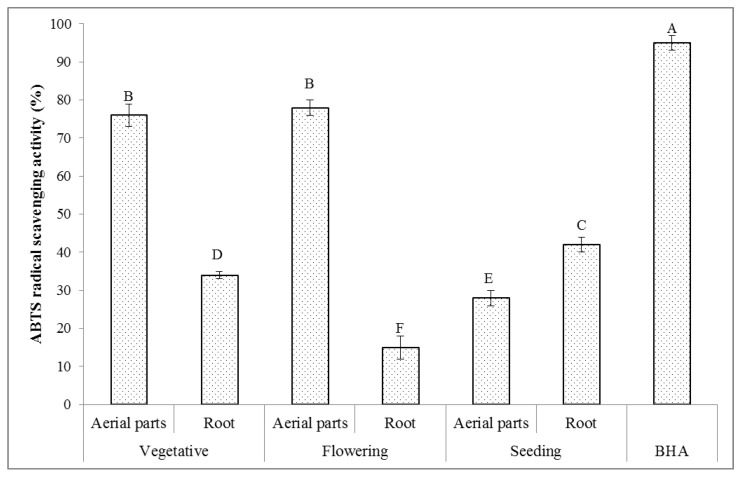
ABTS radical scavenging activity (*n* = 3) of the extracts of the *T. polium* aerial parts and root at different phenological stages. The extract concentration is 200 µg/mL. Various letters indicate statistically significant differences (*p* < 0.05).

**Figure 4 molecules-27-01561-f004:**
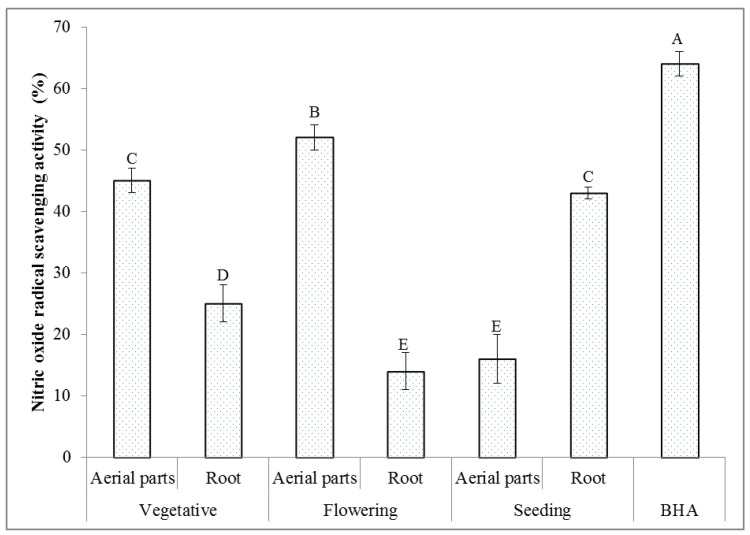
Nitric oxide radical scavenging activity (*n* = 3) of the extracts of the *T. polium* aerial parts and root at different phenological stages. The extract concentration is 200 µg/mL. Various letters indicate statistically significant differences (*p* < 0.05).

**Figure 5 molecules-27-01561-f005:**
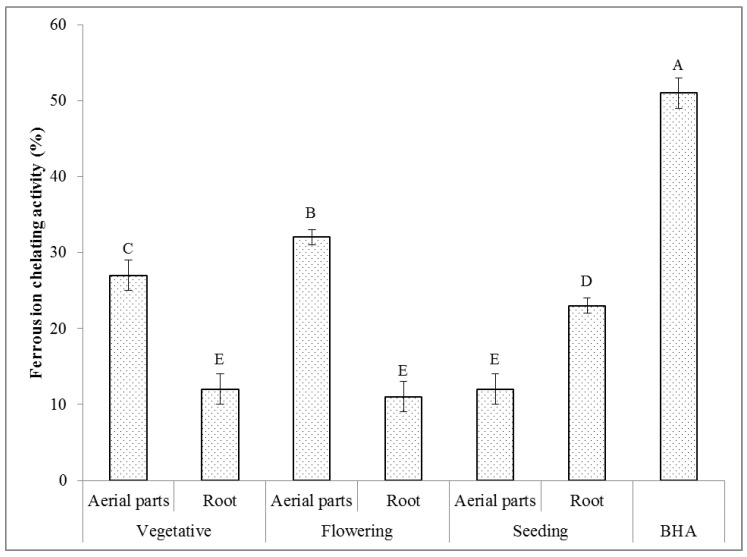
Ferrous ion chelating activity (*n* = 3) of the extracts of the *T. polium* aerial parts and root at different phenological stages. The extract concentration is 200 µg/mL. Various letters indicate statistically significant differences (*p* < 0.05).

**Figure 6 molecules-27-01561-f006:**
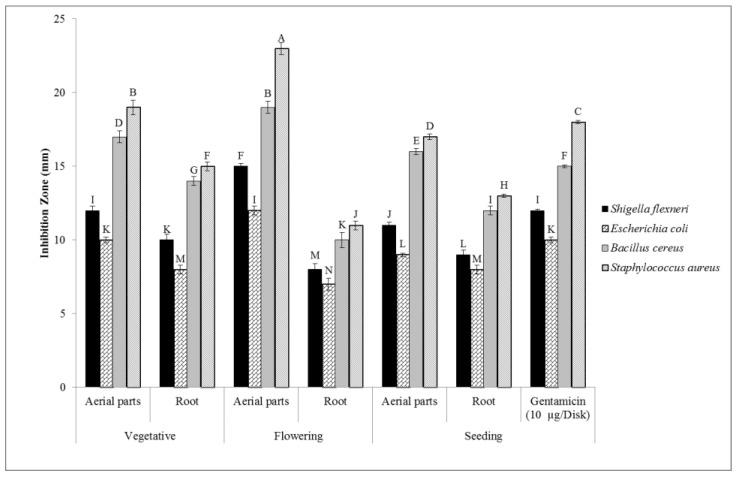
The inhibition zone (*n* = 3) of the extracts of the *T. polium* aerial parts and root at different phenological stages against the Gram-positive (*B. cereus*, *S. aureus*) and Gram-negative bacteria (*Sh. flexneri*, *E. coli*) strains. The extract concentration is 200 µg/mL. Various letters indicate statistically significant differences (*p* < 0.05).

**Figure 7 molecules-27-01561-f007:**
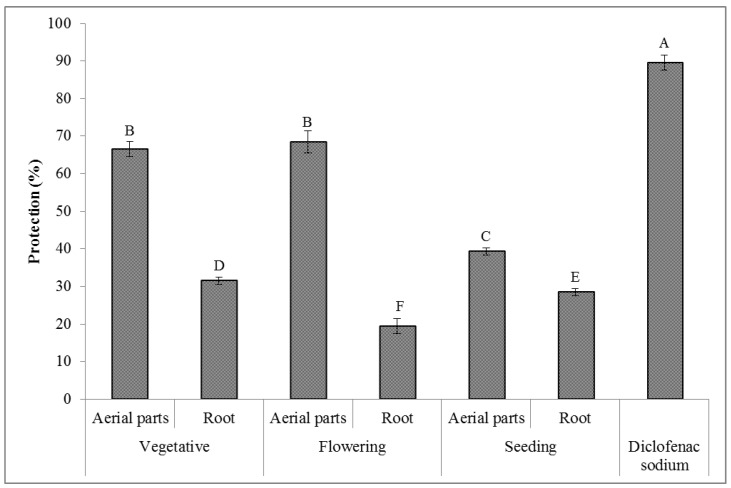
The anti-inflammatory activity (*n* = 3) of the extracts of the *T. polium* aerial parts and root at different phenological stages. The extract concentration is 200 µg/mL. Various letters indicate statistically significant differences (*p* < 0.05).

**Table 1 molecules-27-01561-t001:** The content (*n* = 3) of the volatile hydrocarbons identified and determined in the *T. polium* extracts of the aerial parts and root of this plant collected at different phenological stages. The exemplary GC-MS chromatogram is given in the Appendix A.

				Vegetative		Flowering		Seeding	
	Compound	RI *	Molecular Formula	Aerial Parts (%)	Root (%)	Aerial Parts (%)	Root (%)	Aerial Parts (%)	Root (%)
1	(*E*)-2-Hexenal	860	C_6_H_10_O	0.1 ± 0.02	0.1 ± 0.00	0.1 ± 0.01	-	-	-
2	3-Heptanone	885	C_7_H_14_O	0.1 ± 0.00	-	-	-	-	-
3	α-Thujene	924	C_10_H_16_	0.2 ± 0.00	0.1 ± 0.01	0.4 ± 0.00	0.1 ± 0.00	0.1 ± 0.02	0.1 ± 0.00
4	α-Pinene	935	C_10_H_16_	3.8 ± 0.04	1.9 ± 0.03	4.3 ± 0.00	0.7 ± 0.04	1.3 ± 0.05	2.5 ± 0.03
5	Camphene	950	C_10_H_16_	0.2 ± 0.01	-	0.2 ± 0.00	-	0.1 ± 0.00	-
6	β-Pinene	975	C_10_H_16_	7.2 ± 0.03	4.1 ± 0.05	9.4 ± 0.02	2.5 ± 0.05	3.6 ± 0.03	6.3 ± 0.02
7	1-Octen-3-ol	984	C_8_H_16_O	0.3 ± 0.00	-	-	-	-	-
8	β-Myrcene	993	C_10_H_16_	1.5 ± 0.00	0.5 ± 0.02	1.8 ± 0.03	0.2 ± 0.01	0.3 ± 0.00	0.8 ± 0.02
9	α-Phellandrene	1006	C_10_H_16_	-	-	0.1 ± 0.00	0.1 ± 0.00	-	-
10	α-Terpinene	1015	C_10_H_16_	0.3 ± 0.01	-	-	-	-	-
11	p-Cymene	1025	C_10_H_14_	0.4 ± 0.00	0.1 ± 0.00	0.7 ± 0.02	0.1 ± 0.00	-	0.1 ± 0.01
12	Limonene	1030	C_10_H_16_	3.6 ± 0.02	1.7 ± 0.01	4.2 ± 0.00	0.6 ± 0.03	1.0 ± 0.02	2.8 ± 0.06
13	1.8-Cineole	1034	C_10_H_18_O	0.2 ± 0.00	0.1 ± 0.00	0.3 ± 0.01	0.1 ± 0.02	-	-
14	*cis*-β-Ocimene	1041	C_10_H_16_	-	0.1 ± 0.01	-	0.1 ± 0.00	-	-
15	*trans*-β-Ocimene	1049	C_10_H_16_	0.1 ± 0.00	0.1 ± 0.00	-	-	-	-
16	γ-Terpinene	1062	C_10_H_16_	0.3 ± 0.01	-	0.5 ± 0.02	0.2 ± 0.00	0.2 ± 0.01	0.1 ± 0.00
17	Terpinolene	1094	C_10_H_16_	-	-	0.2 ± 0.00	-	-	-
18	Linalool	1105	C_10_H_18_O	0.5 ± 0.02	0.1 ± 0.01	0.6 ± 0.00	0.2 ± 0.01	0.1 ± 0.00	0.2 ± 0.00
19	4-Acetyl-1-methylcyclohexene	1118	C_9_H_14_O	0.1 ± 0.00	-	0.1 ± 0.01	-	-	-
20	*trans*-Pinocarveol	1125	C_10_H_16_O	1.5 ± 0.02	0.2 ± 0.00	0.1 ± 0.00	0.1 ± 0.02	0.1 ± 0.00	-
21	Camphor	1145	C_10_H_16_O	0.3 ± 0.00	0.1 ± 0.02	0.4 ± 0.03	-	-	0.3 ± 0.00
22	Pinocarvone	1160	C_10_H_14_O	0.4 ± 0.02	0.1 ± 0.02	0.6 ± 0.00	0.1 ± 0.03	0.1 ± 0.00	0.2 ± 0.01
23	Borneol	1168	C_10_H_18_O	0.3 ± 0.00	0.2 ± 0.00	-	-	-	-
24	Terpinen-4-ol	1178	C_10_H_18_O	0.2 ± 0.01	0.1 ± 0.00	0.5 ± 0.03	-	-	0.2 ± 0.00
25	p-Cymen-8-ol	1188	C_10_H_14_O	-	-	0.1 ± 0.00	0.1 ± 0.02	0.1 ± 0.00	-
26	α-Terpineol	1195	C_10_H_18_O	0.1 ± 0.00	-	0.1 ± 0.00	-	0.1 ± 0.01	-
27	Myrtenol	1197	C_10_H_16_O	0.5 ± 0.01	0.2 ± 0.01	0.7 ± 0.04	-	-	0.2 ± 0.00
28	Verbenone	1204	C_10_H_14_O	-	0.1 ± 0.00	0.1 ± 0.00	0.1 ± 0.01	-	0.1 ± 0.00
29	Cuminaldehyde	1215	C_10_H_12_O	-	0.1 ± 0.00	-	0.1 ± 0.01	-	-
30	Bornyl acetate	1285	C_12_H_20_O_2_	0.1 ± 0.00	-	0.2 ± 0.00	0.1 ± 0.02	0.1 ± 0.00	0.1 ± 0.01
31	α-Fenchyl acetate	1294	C_12_H_20_O_2_	0.1 ± 0.00	-	-	-	-	-
32	Thymol	1305	C_10_H_14_O	0.2 ± 0.01	-	0.4 ± 0.03	-	0.1 ± 0.00	-
33	Carvacrol	1319	C_10_H_14_O	6.2 ± 0.05	4.5 ± 0.03	8.2 ± 0.00	2.3 ± 0.05	3.8 ± 0.06	5.6 ± 0.02
34	α-Cubebene	1349	C_15_H_24_	0.3 ± 0.00	0.2 ± 0.01	-	-	-	-
35	α-Copaene	1374	C_15_H_24_	0.8 ± 0.03	0.2 ± 0.00	1.1 ± 0.02	0.1 ± 0.00	0.2 ± 0.00	0.5 ± 0.03
36	β-Bourbonene	1380	C_15_H_24_	0.6 ± 0.01	0.1 ± 0.00	0.9 ± 0.04	0.1 ± 0.00	-	0.4 ± 0.05
37	β-Elemene	1395	C_15_H_24_	0.1 ± 0.00	-	0.2 ± 0.01	-	0.1 ± 0.00	0.1 ± 0.01
38	β-Caryophyllene	1415	C_15_H_24_	0.5 ± 0.00	0.2 ± 0.01	0.7 ± 0.03	0.1 ± 0.00	0.1 ± 0.02	-
39	α-Bergamotene	1435	C_15_H_24_	-	-	0.1 ± 0.00	0.1 ± 0.00	-	-
40	α-Humulene	1454	C_15_H_24_	0.1 ± 0.00	-	0.1 ± 0.01	-	0.1 ± 0.02	-
41	Germacrene D	1478	C_15_H_24_	15.2 ± 0.07	11.3 ± 0.05	17.4 ± 0.08	7.6 ± 0.04	9.4 ± 0.06	13.4 ± 0.05
42	Bicyclogermacrene	1494	C_15_H_24_	6.3 ± 0.05	4.1 ± 0.00	7.2 ± 0.05	2.1 ± 0.03	3.2 ± 0.02	5.5 ± 0.01
43	γ-Cadinene	1512	C_15_H_24_	1.8 ± 0.04	0.8 ± 0.01	2.1 ± 0.03	0.3 ± 0.04	0.5 ± 0.02	1.4 ± 0.00
44	δ-Cadinene	1519	C_15_H_24_	1.2 ± 0.01	0.4 ± 0.03	2.5 ± 0.02	0.1 ± 0.00	0.2 ± 0.01	0.8 ± 0.03
45	α-Calacorene	1530	C_15_H_20_	0.1 ± 0.00	0.1 ± 0.00	-	-	-	-
46	Spathulenol	1571	C_15_H_24_O	0.7 ± 0.03	0.2 ± 0.00	1.5 ± 0.01	0.1 ± 0.00	0.1 ± 0.02	0.5 ± 0.00
47	Viridiflorol	1588	C_15_H_26_O	1.4 ± 0.06	0.6 ± 0.03	2.0 ± 0.04	0.2 ± 0.00	0.5 ± 0.02	0.9 ± 0.00
48	t-Cadinol	1638	C_15_H_26_O	12.3 ± 0.02	9.1 ± 0.04	14.5 ± 0.05	5.1 ± 0.03	7.5 ± 0.04	10.6 ± 0.00
49	Nootkatone	1775	C_15_H_22_O	0.1 ± 0.00	-	0.1 ± 0.00	-	0.1 ± 0.00	-
Total identified compounds %				70.3	41.8	84.7	23.7	33.1	53.7

* RI—retention index.

**Table 2 molecules-27-01561-t002:** The minimum inhibitory concentrations (MICs) and the minimum bactericidal concentrations (MBCs) of the extracts of the *T. polium* aerial parts and root at different phenological stages for the Gram-positive (*B. cereus*, *S. aureus*) and Gram-negative (*Sh. flexneri*, *E. coli*) bacteria.

Phenological Stages	Plant Parts	*Shigella flexneri*		*Escherichia coli*		*Bacillus cereus*		*Staphylococcus aureus*	
		MIC (µg/mL)	MBC (µg/mL)	MIC (µg/mL)	MBC (µg/mL)	MIC (µg/mL)	MBC (µg/mL)	MIC (µg/mL)	MBC (µg/mL)
**Vegetative**	Aerial parts	37.5	75	75	150	18.75	37.5	18.75	37.5
	Root	75	150	150	300	37.5	75	37.5	75
**Flowering**	Aerial parts	18.75	37.5	37.5	75	9.4	18.75	9.4	18.75
	Root	300	600	300	600	150	300	150	300
**Seeding**	Aerial parts	75	150	150	300	37.5	75	18.75	37.5
	Root	150	300	300	600	75	150	75	150

## Data Availability

The data presented in this study are available on request from the corresponding author.

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
