# Peer review of "Teucrium polium (L.): Phytochemical Screening and Biological Activities at Different Phenological Stages"

_molecules, 2022, doi:10.3390/molecules27051561_

Round 1
Reviewer 1 Report
The article "Teucrium polium (L.): Phytochemical Screening and Biological Activities at Different Phenological Stages" by Majid Sharifi-Rad, Pawel Pohl, Francesco Epifano, Gokhan Zengin, Nidal Jaradat and Mohammed Messaoudi is of interest due to the increasing role of medicinal plant components as alternatives to traditional chemical preparations in the treatment of various diseases and pathological conditions. Therefore, the topic is relevant.
The authors made a large amount of different experiments, but the article is descriptive and superficial. The analysis of the data was not carried out.
- The novelty of the study is not clear, a large number of studies of this object are present in the literature.
- The choice of extractant is not entirely clear. It is clear that methanol is optimal in terms of completeness of extraction. But how to use such extracts as a biologically active food supplement in the future?
- The optimal extraction parameters were not identified, the extraction was carried out in the only way.
- Fig1. The results obtained using different approaches are significantly correlated, although the nature of the studied compounds is quite different. It makes sense to consider in more detail the chemistry of the methods that were used. In particular, the Total Flavonoids Content method is not selective for the flavonoid group, as it is based on the use of their complexing properties. Therefore, many polyphenolic compounds with hydroxyl groups in the ortho- and, in some cases, meta-position can enter into the reaction of complexation with aluminum.
- 2.6.1-2.6.4. There is no analysis of the obtained data. In fact, all the methods, given in these paragraphs, reflect the antioxidant properties of the compounds. There is no discussion of mechanisms; there is no discussion of correlations. In principle, there is not a single chemical reaction in the text of the manuscript.
- The conclusion is very superficial; the results should be worked out more deeply.
The manuscript requires major revision for publication in the Molecules.
Author Response
Please, see the attachment.

Reviewer 2 Report
Though the topic is important considering the need of natural products in various therapies, it requires substantial and thorough revision with additional information in order to appreciate the quality.
Please reduce the content of the first paragraph of the introduction, as it is providing very basic and general information. Instead increase the content and strengthen the background data on T. polium as folk medicine in Iran. Provide more information on T. polium, its fundamental details and traditional importance. You can see below latest references on T. polium, which can be cited for better impact and help in providing data on T. polium folk medicine importance and directly relevant with the work
https://pubmed.ncbi.nlm.nih.gov/33167507/
https://pubmed.ncbi.nlm.nih.gov/33114026/
In all the bar graphs, I suggest to replace alphabets with Asterisk symbol for showing significance. Alphabets are creating a confusion while analyzing the results.
The sequence of presenting results in all bar graphs is completely wrong. Therefore, it is difficult to follow and compare the results properly. First bar should be seeding stage, followed by vegetative stage and then flowering stage. This sequence will provide the right presentation. please change all graphs accordingly.
If I am not wrong, it should be seedling and not seeding. Please confirm.
Where is the chromatogram? please add GC chromatogram in the manuscript to confirm the compounds as well as mass spectra peaks for all identified compounds as supplementary file.
Where is the control in Antibacterial assay. There is no control in figure 6. Please add control data. Authors must understand the importance of control. Without control, how you gonna compare? Standard rejection of manuscripts straight is due to the missing control.
Please dont use full microbial names everywhere in the manuscript. Bacillus cereus, Staphylococcus aureus, Shigella flexneri, Escherichia coli...........Use only once as it appears first with abbreviated form in parenthesis. Followed by abbreviated name throughout the manuscript.
Please provide the latitude and longitude coordinates for exact plant collection sites.
The biggest problem in this manuscript is lack of proper discussion. No comparative analysis is properly done with previous studies. For example I will provide one instance in section 2.4 total saponin content. Authors states "These results are also in accordance with the previous findings reported in literature for other plants [22, 23]". However, if we look at the reference number 23, it is a review article. It is not a study to compare with. Similarly, reference 22 is about Cyclocarya paliurus. How authors can just simply say that results are also in accordance with the previous findings. It looks like that similar study was done on T. polium. But this is not the case. Throughout the manuscript in the discussion, authors must clearly refer write medicinal plant in the sentence to have the right information and not misleading. It should be revised like "Results are also in accordance with the previous findings with Cyclocarya paliurus"....etc.
Therefore, I suggest authors to thoroughly check all the references and strengthen your discussion with more comparative data and mention all plants name.
Author Response
Please, see the attachment.

Round 2
Reviewer 1 Report
Most of the comments have been taken into account, the article can be accepted for publication
Author Response
Replies to the Reviewer 1, stage 2
Reviewer 2 Report
Manuscript is significantly improved by the authors. However, there are still some minor concerns. Introduction is unnecessarily long and only about medicinal plants. I requested authors to strengthen the introduction with T. polium importance in folk medicine. Delete some parts from start and add further data about T. polium. Similarly, discussion section still needs improvement. I could not see any significant improvement in discussion.
Author Response
Manuscript is significantly improved by the authors. However, there are still some minor concerns. Introduction is unnecessarily long and only about medicinal plants. I requested authors to strengthen the introduction with T. polium importance in folk medicine. Delete some parts from start and add further data about T. polium. Similarly, discussion section still needs improvement. I could not see any significant improvement in discussion.
Answer:
The manuscript has been improved following the suggestions of the reviewer about the introduction and the discussion sections.